# Wear of Carbide Plates with Diamond-like and Micro-Nano Polycrystalline Diamond Coatings during Interrupted Cutting of Composite Alloy Al/SiC

Evgeny E. Ashkinazi [1], Sergey V. Fedorov [2,*], Artem K. Martyanov [1], Vadim S. Sedov [1], Roman A. Khmelnitsky [3], Victor G. Ralchenko [1], Stanislav G. Ryzhkov [1], Andrey A. Khomich [4], Mikhail A. Mosyanov [2], Sergey N. Grigoriev [2] and Vitaly I. Konov [1]

[1] Department of Light-Induced Surface Phenomena, Prokhorov General Physics Institute of the Russian Academy of Sciences, Vavilov Str. 38, 119991 Moscow, Russia; jane50@list.ru (E.E.A.); art.martyanov@gmail.com (A.K.M.); sedovvadim@yandex.ru (V.S.S.); vg_ralchenko@mail.ru (V.G.R.); high-low@yandex.ru (S.G.R.); vitaly.konov@gmail.com (V.I.K.)
[2] Department of High-Efficiency Machining Technologies, Moscow State University of Technology STANKIN, Vadkovskiy Per. 3A, 127055 Moscow, Russia; mmosyanov@yandex.ru (M.A.M.); s.grigoriev@stankin.ru (S.N.G.)
[3] Laboratory of Physics of Defects and Phase Transformations in Solid Structures, Division of Solid-State Physics, P. N. Lebedev Physical Institute of the Russian Academy of Sciences, Leninsky Avenue, 53, 119991 Moscow, Russia; khmelnitskyra@lebedev.ru
[4] Semiconductor Growth Laboratory MOCVD, Fryazinsky Branch of the V. A. Kotelnikov Institute of Radio Engineering and Electronics of the Russian Academy of Sciences, 1 Vvedensky Square, 141190 Fryazino, Russia; antares-610@yandex.ru
* Correspondence: sv.fedorov@stankin.ru; Tel.: +7-916-290-26-07

**Abstract:** The complexity of milling metal matrix composite alloys based on aluminum like Al/SiC is due to their low melting point and high abrasive ability, which causes increased wear of carbide tools. One of the effective ways to improve its reliability and service life is to modify the surface by plasma chemical deposition of carbon-based multilayer functional layers from vapor (CVD) with high hardness and thermal conductivity: diamond-like (DLC) or polycrystalline diamond (PCD) coatings. Experiments on an indexable mill with CoroMill 200 inserts have shown that initial tool life increases up to 100% for cases with DLC and up to 300% for multilayered MCD/NCD films at a cutting speed of 800 m/min. The primary mechanism of wear of a carbide tool in this cutting mode was soft abrasion, when wear on both the rake and flank surfaces occurred due to the extrusion of cobalt binder between tungsten carbide grains, followed by their loss. Analysis of the wear pattern of plates with DLC and MCD/NCD coatings showed that abrasive wear begins to prevail against the background of soft abrasion. Adhesive wear is also present to a lesser extent, but there is no chipping of the base material from the cutting edge.

**Keywords:** Al/SiC composite alloy; milling; tungsten carbide; diamond-like coating; micro-nanocrystalline multilayered diamond coating; wear

## 1. Introduction

Aluminum alloys and composite materials based on them, due to their high specific strength and rigidity, are widely used as the main components of internal combustion engines, load-bearing elements in aircrafts, and rocket engineering [1], for which weight reduction, an increase in elastic modulus, and the strength-to-weight ratio are most important [2].

Composites with a metal matrix (MMC) made of aluminum alloy reinforced with dispersed particles have been actively developed over the past decade by modern technological requirements, as well as consumer specifications for newly developed systems and machines [3]. They represent a relatively new class of heterophase structural materials,

which is actively developing worldwide and is increasingly being used in existing and promising products in various fields of industrial production. MMCs are manufactured by a highly efficient mixing casting method in which the required amount of hot filler in powder or fiber is loaded into an aluminum alloy metal, melted, and stirred as the system temperature decreases for several minutes [4].

It was found that the presence of such particles as SiC, $Al_2O_3$, and $B_4C$ in composites improved hardness, density, tensile strength, and tribological characteristics, provided a higher modulus of elasticity, a lower coefficient of thermal expansion, and a higher hardness and wear resistance of the composite [5,6]. In particular, adding a moderate volume fraction of SiC to an aluminum alloy leads only to a slight increase in density since it is practically the same for both components. Thus, such composites can directly replace aluminum alloys without a significant increase in weight.

Traditional machining of aluminum and its alloys is not complicated. One of the main problems preventing the widespread use of alloys with a high SiC content in mechanical engineering is the high production costs that arise during the machining of these materials. Chip control, compliance with strict dimensional tolerances, good surface finish, and ensuring minimal warping can become big problems [7]. The latter is a consequence of the high plasticity of this material and its tendency to stick to the surface of cutting tools.

The reinforcement of an aluminum alloy complicates the technology of its processing since the tool encounters an inhomogeneous material on its way [8]. The high hardness of SiC particles makes them difficult to process even in dry cutting conditions, and this negatively affects both carbide tools and processed surfaces, causing their accelerated wear on the surface to be processed [9]. Moreover, in combination with high mechanical strength, a percentage of SiC varying from 10 to 20%, and high cutting speeds, the cutting temperature tends to increase sharply since solid particles create intermittent friction on the tool's surface. Moreover, the cutting speed always has the most significant effect [10]. The processing temperature can exceed 700 °C, accelerating the mechanisms of adhesive and abrasive wear, but it has little impact on the cutting force [11]. An increase in the feed rate, provided that this does not lead to an excessive rise in the effective chip-tool contact area, leads to more excellent heat dissipation on the tool–workpiece interface and also contributes to an increase in temperature. However, in most cases, the low melting point of aluminum and its alloys causes a relatively low temperature in the cutting zone, which, in most cases, prevents the development of thermally activated wear mechanisms, such as diffusion wear. However, high temperatures arising during high-speed milling can still start this due to the penetration of aluminum into the tool material and the formation of cobalt aluminide (AlCo), which reduces the bond strength between carbide grains, which can lead to chipping of the cutting edge [12].

One of the factors limiting the feed rate and cutting depth may be the surface roughness, which is also influenced by the hardness and characteristics of the microstructure. The higher the hardness of the processed alloys, the lower the roughness of their surface due to the reduction of sticking to the tool [7]. However, when the hardness results from the introduction of solid particles into the aluminum matrix, solid particles may accidentally fall out of the matrix, which will scratch it. In addition, the high chemical affinity of aluminum alloys to tool materials leads to sticking on the tool's surface, which also negatively affects the quality of the character.

However, high roughness values can be minimized using cutting tools with a low chemical affinity for aluminum, coolant, large front corners, low feed rates, and a larger radius of rounding at the top [13,14]. These conditions facilitate chip runoff and prevent the formation of sticking. However, it should be borne in mind that a high SiC content can increase surface roughness at higher cutting speeds due to a rapid increase in wear on the back surface. The addition of up to 0.5 wt% tin, lead, or bismuth in combination with high cutting speeds reduces the value of surface roughness since it makes the chips brittle and facilitates sliding.

Cutting tools based on natural, PCD, or CVD diamonds are offered for the mechanical processing of aluminum matrix composites, which, unfortunately, is quite limited due to their high price. Polycrystalline diamond (PCD) tools are more suitable for processing Al/SiC alloys because their hardness is three to four times higher than ceramic particles such as SiC. Moreover, the thermal conductivity of PCD tools is about four times higher than that of carbide tools, which has a favorable effect on the cutting temperature.

It should be noted that alternative methods of removing aluminum matrix composite materials are rapidly developing using electrical discharge treatment, abrasive water jet treatment, and laser technology [15,16], or processes related to additive technologies are used [17].

One of the ways to increase the wear resistance of traditional carbide tools is to apply carbon-based reinforcing coatings, in particular, diamond-like (DLC) [18,19] or PCD [20], to the cutting wedge. Such a tool shows the lowest level of wear, although sometimes it does not provide the best roughness. However, milling cutters with DLC are much more widespread due to the complex multi-stage technology for obtaining a diamond-based coating, which determines its relatively high price.

The purpose and novelty of this work was a comparative study of machinability when milling an Al/SiC composite with tungsten carbide plates with wear-resistant carbon-based coatings obtained using various technologies: CVD, PCD, and DLC deposited on a composite nanocrystalline coating based on (TiCrAlSi)N.

## 2. Materials and Methods

### 2.1. Cutting Tools and Machined Material

The intermittent cutting research was conducted on a CoroMill 200 indexable mill with a Sandvik Coromant cylindrical shank (Sandviken, Sweden). Two round-shaped inserts made of hard alloy H10F with a submicron carbide grain size of 0.8 microns, cutting group S, were installed. The choice of these plates is explained by the fact that they are recommended to be used, including for hard-to-process materials such as nickel and titanium alloys, due to their high ability to resist abrasive and adhesive wear. The geometric parameters of the milling cutter with installed inserts, controlled by a Helicheck Plus machine (Walther, Lp, Tübingen, Germany), are shown in Table 1.

**Table 1.** The mill geometrical parameters.

| Parameter | Value |
|---|---|
| Diameter | 24.886 mm |
| Number of teeth | 2 |
| End teeth runout | 0.0013 mm |
| Insert radius | 5.0398 mm |
| Spiral angle (spiral left) | 13.971° |
| The rake angle is 1 mm from the top | 3.209° |
| The flank angle is 1 mm from the top | 18.805° |

The inserts mounted on the mill were tested on a CTX beta 1250 TC lathe (DMG MORI (Bielefeld, Germany) equipped with a Siemens CNC system (Munich, Germany)). The cutting scheme is shown in Figure 1.

Accelerated tests with passing milling were car-ried out along the end of the cylindrical work-piece in a spiral. The cutting mode was as follows: the cutting speed v wais 800 m/min, the feed per tooth Fz wais 0.2 mm, the cutting depth t wais 1 mm, and the milling width wais 12 mm. The pro-cessing time for one pass of the cutter (complete processing of the end surface of the workpiece) was 20 s, corresponded to the cutting length of 0.4 m with each insert.

A rod from MMC A390 with a SiC phase content of 18% with a diameter of 75 mm was selected as the workpiece, the chemical composition of which, carried out on the Q4

TASMAN spark optical emission spectrometer (Bruker, Billerica, MA, USA), is shown in Table 2. The structure of the composite alloy is shown in Figure 2.

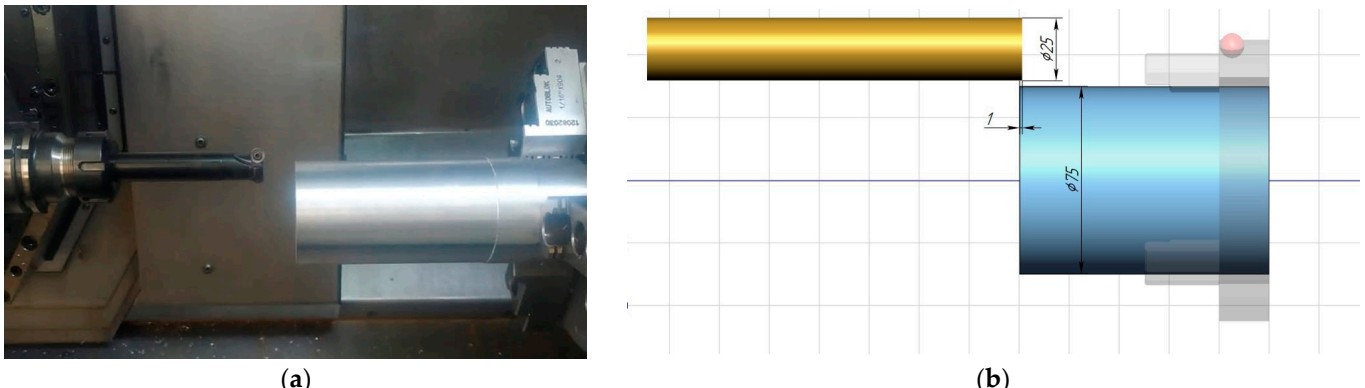

|  | (a) |  | (b) |

**Figure 1.** The milling test on mills; the workpiece installed on the machine (**a**), processing scheme (**b**).

**Table 2.** Chemical composition of the MMC A390 billet (wt%).

| Si | Fe | Cu | Mn | Mg | Cr | Ni | Zn | Ti | Bi |
|---|---|---|---|---|---|---|---|---|---|
| 13.49 | 1.032 | 4.228 | 0.021 | 0.506 | 0.024 | 0.012 | 0.012 | 0.036 | 0.033 |
| Ga | Li | Na | P | Pb | Sn | V | Zr | Sb | Al |
| 0.017 | 0.013 | 0.016 | 0.005 | 0.128 | 0.020 | 0.049 | 0.019 | 0.009 | Base |

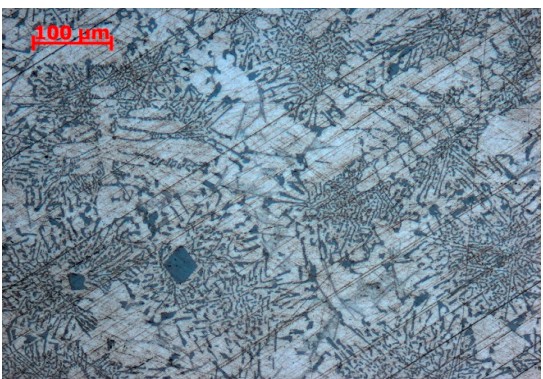

**Figure 2.** Metallography of A390 alloy workpiece.

### 2.2. Tool Coatings

The well-proven CVD technologies of diamond-like and polycrystalline diamond coatings were selected when applying carbon films to carbide plates.

Before applying a diamond coating, it is necessary to pay attention to the condition of the substrate, which was an alloy H10F containing 10% Co. However, X-ray spectral analysis from the surface (JSM-7001F, JEOL, Tokyo, Japan) showed its significant enrichment with cobalt up to 75% (Figure 3a,b), probably due to the peculiarities of the plate manufacturing technology.

Direct deposition with good adhesion of a high-quality diamond-like or diamond film onto plates in this state is complex due to the catalytic effect of Co, which contributes to the formation of graphite. Therefore, the inserts were subjected to intensive etching with argon and chromium ions before applying the DLC. For the case of PCD coating, the chemical etching was performed with Murakami reagent ($K_3Fe(CN)_6$:KOH:$H_2O$ = 1:1:10) for 10 min and then 4 s with Caro acid ($H_2SO_4$-$H_2O_2$ solution). Then, a barrier layer of tungsten with preliminary ionic assistance was applied to the cutting plates by magnetron sputtering to form a layer up to 0.5 μm thick at the boundary between the substrate and the coating,

preventing undesirable interaction of carbon and cobalt, and allowing to reduce residual thermal stresses due to almost identical thermal expansion coefficients of tungsten carbide and hard alloy [21,22].

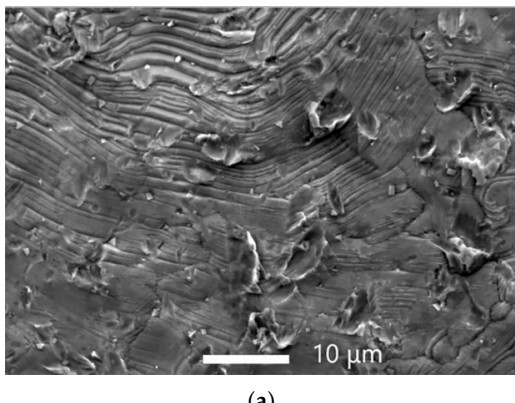

(**a**)

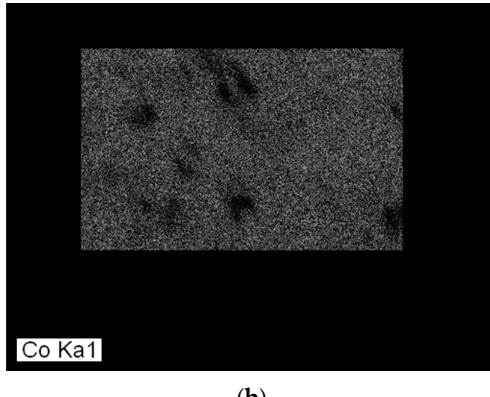

(**b**)

**Figure 3.** SEM image of the initial insert surface (**a**); Co distribution on the plate surface (**b**).

Diamond-like coatings have a lower coefficient of friction on cutting surfaces than nitride coatings. However, their use in cutting hard-to-process materials is limited by low heat resistance and an increased level of internal stresses, which affects the strength of the adhesive joint with the substrate [23]. Significant progress in improving a coating is associated with its alloying, particularly with silicon [24], which leads to a restructuring of the structure and contributes to the formation of secondary structures that increase wear resistance [25]. The application of DLC to the transition nitride layer of ncAlTiCrN/Si$_3$N$_4$ compounds with a nano-structural component resistant to abrasive wear showed high efficiency [26–28].

The ncAlTiCrN/Si$_3$N$_4$ + DLC (a-C:H:Si type) coating was applied on the Platit π311 + DLC installation (Platit, Grenchen, Switzerland). The application time of the nitride layer ncAlTiCrN/Si$_3$N$_4$ with a thickness of 2.5–2.9 μm was 1 h. They were obtained using the PVD method from cylindrical rotating cathodes Ti, AlSi18%, and Cr. This class of nanocomposite coatings was specially developed to counteract high-temperature wear [29]. Their unique properties are provided by the presence in the structure of at least two phases: nanocrystalline, consisting of crystallites (TiCrAlSi)N with a size of about 5 nm, and an amorphous Si$_3$N$_4$ matrix [30].

It took 2 h to form a diamond-like coating based on a-C:H:Si by PACVD with a thickness of 1 μm. Figure 4 shows the grinding image of the coating (Calowear, GFM, Graz, Austria) and the SEM image of the coating (Vega 3, Tescan, Brno, Czech Republic), which shows that the film consists of fused spherulites ranging in size from 0.2 to 2.5 μm.

Polycrystalline diamond coating with alternating layers with micro- and nanostructure (MCD/NCD) was grown in a microwave (2.45 GHz) plasma CVD reactor ARDIS-100 (Optosystems, Moscow, Russia) [31,32]. The lower layer of the coating was grown in a mode that ensures the growth of the microcrystalline diamond fraction for 10 min for the reliable formation of diamond nuclei from the deposited layers of diamond particles seed [33] in a methane/hydrogen gas mixture with a methane concentration of 4% at a substrate temperature of 850 °C. Then, a multilayer polycrystalline diamond film was formed by periodic nitrogen injection, which ensures the growth of the nanocrystalline fraction [34], reducing the overall roughness of the coating surface. The total deposition time was 6 h with an average growth rate of ~1 μm/h. The structure of the coating surface is shown in Figure 5. In addition, the near-surface layer of a solid with a thickness of 24 μm, modified as a result of chemical etching, is visible on the fracture of the plate.

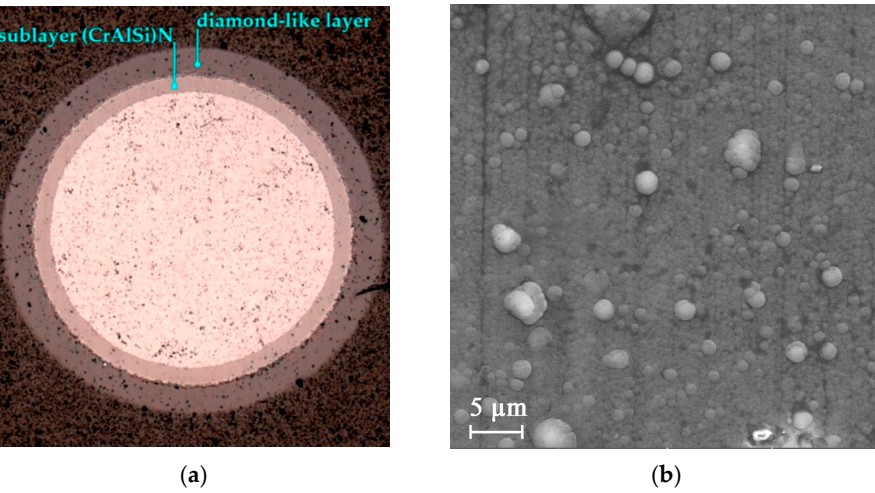

(**a**)    (**b**)

**Figure 4.** (**a**) Grinding image of ncAlTiCrN/Si$_3$N$_4$ + DLC coating, (**b**) SEM image of DLC film.

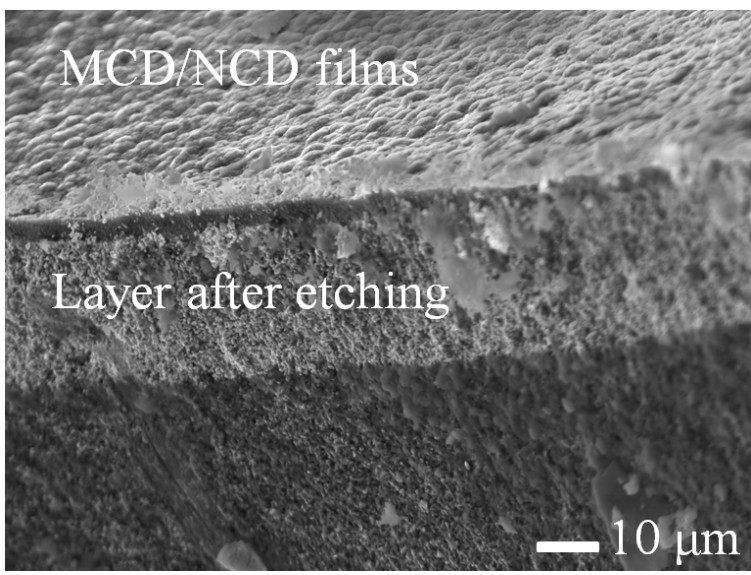

**Figure 5.** The cutting plate fracture has a diamond (MCD/NCD) coating and a modified chemical etching near-surface layer.

Structural phase analysis of DLC and MCD/NCD coatings was performed by Raman spectroscopy using a LabRam HR800 spectrometer (HORIBA, Kyoto, Japan) with the following parameters: laser wavelength 473 nm, power 10 mW. The beam was focused with an Olympus × 100 long-focus lens (NA = 0.9) into a spot with a diameter of 1 μm. When measuring the lines of the Raman spectra, a diffraction grating with a stroke density of 1800 mm$^{-1}$ was used.

The Raman spectra for samples of the CoroMill-200 inserts with two types of carbon coatings were recorded at a distance of 0.5 mm from the cutting age. The Raman spectra in Figure 6a are typical for nanocrystalline CVD diamond film and DLC film.

The spectrum of the DLC film is characteristic of amorphous sp$^2$ carbon (a-C) [35]. Such a material has an oscillation mode at 1350 cm$^{-1}$ (D-peak), which is responsible for the respiratory vibrations of six-membered carbon rings [36], and the 1580 cm$^{-1}$ (G-peak) mode corresponds to symmetric stretching of sp$^2$ bonds. There is no diamond peak on the spectrum of the DLC film. Nevertheless, a study conducted on the K-Alpha X-ray spectrometer (Thermo Fisher Scientific, Waltham, MA, USA) showed the presence of a significant content of amorphous carbon of sp$^3$ bond at a level of 14% (Figure 6b).

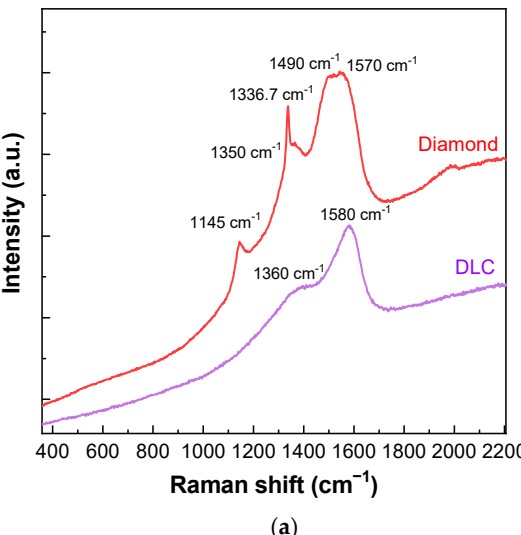

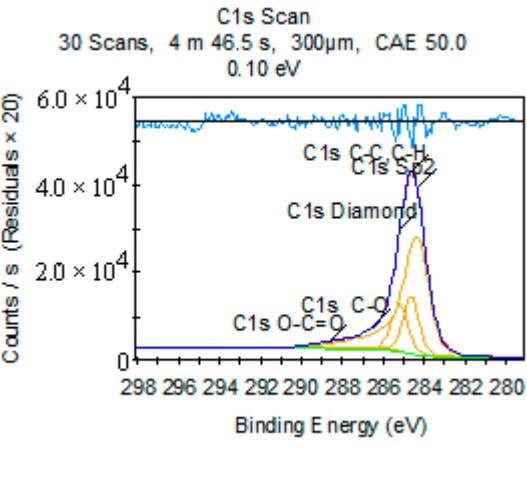

(**a**)  (**b**)

**Figure 6.** Raman spectra of CVD and DLC coating samples at a distance of 0.5 mm from the cutting edge (**a**); XPS profile of the chemical state of carbon in the DLC coating (**b**).

Diamond spectra are characteristic of single-crystal CVD diamonds. During plasma chemical deposition on the insert, the film was expanded due to heating, and when cooled after synthesis, it decreased in size, so the diamond coating was compressed. The diamond peak is located at 1336.7 cm$^{-1}$, which indicates the presence of an elastic compression stress of ~2 GPa. The width of the diamond peak is 10.4 cm$^{-1}$. This value is typical for NCD films. Also, in the spectra, there are bands at 1145 and 1490 cm$^{-1}$ associated with trans polyacetylene—chains of C-H bonds at the film grain boundaries. The bands at 1350 (D) and 1570 (G) cm$^{-1}$ refer to amorphous sp$^2$ carbon. The entire ensemble of bands is typical for NCD films [37].

The comparison of carbon coatings is summarized in Table 3.

**Table 3.** The comparison of diamond-like and polycrystalline diamond coatings.

|  | **Diamond-like Coating** | **Polycrystalline Diamond Coating** |
|---|---|---|
| Type | A 1 μm thickness amorphous carbon film a-C:H:Si is grown on a 2.6 μm intermediate nitride ncAlTiCrN/Si$_3$N$_4$ layer. | A 6 μm thickness 12-layer diamond coating with alternating crystals with micro- and nanostructure (MCD/NCD). |
| Technique | The plasma-enhanced chemical vapor deposition using acetylene (C$_2$H$_2$) and tetramethyl silane (Si(CH$_3$)$_4$) on Platit $\pi$311 + DLC installation. | The microwave (2.45 GHz) plasma CVD reactor ARDIS-100 uses a methane/hydrogen gas mixture with periodic nitrogen injection. |
| Structure | The surface microstructure of DLC-Si coatings is globular. The diameter of the globules does not exceed 2 μm, but droplets with a diameter of 3–5 μm were formed due to the deposition technique. | Microcrystalline diamond layers consist mainly of grains with a size of 1–2 μm. However, the size of individual crystallites can reach 10 μm. Nanocrystalline particles consist primarily in randomly oriented grains with a size of no more than 500 nm. |
| Properties | HIT25 = 25.4 GPa EIT = 328 GPa HV = 2318 Vickers | HIT25 = 28.5 GPa EIT = 719 GPa HV = 2601 Vickers |

## 3. Results and Discussion

With the improvement of tool production technology, the scope of application of tools made of hard alloys is expanding. Traditional and new technological processes, such as high-tech physical–chemical methods of surface hardening, are used to significantly increase the wear resistance and strength of carbide cutters. The choice of the method of surface modification of the tool material, conditions, and parameters of its implementation

to improve operational properties is a multifactorial task, in which it is essential to solve the issue of rational combination of the operating modes of the cutter with the methods of its hardening surface treatment, creating an optimal set of surface properties involved in contact interaction during cutting. The most effective way to form the properties of the tool material necessary in specific conditions is the use of modern methods of surface modification of material properties. Such strategies include the application of wear-resistant coatings by physical and chemical vapor deposition, the difference between which lies only in the form of delivering components to the working area of the vacuum chamber. In modern installations, it is often possible to implement both ways.

In the case of coating deposition processes, interdiffusion reactions between the tool material and the condensate make a significant contribution, and make possible the creation of multilayer coatings consisting of compounds of titanium, aluminum, chromium, silicon, and other elements on the surface of the tool material. Carbon diamond-like and diamond coatings occupy an important place here.

The main problem of efficient aluminum processing is to achieve the maximum material removal speed without destroying the tool. Aluminum can melt and stick to the instrument under extreme heat. Composites based on it with a metal matrix are known for their high abrasiveness. At the same time, using coatings intended for processing steels is not the best idea. A polished, uncoated milling cutter may work better.

Using a coolant, in the case of processing aluminum-based materials on a carbon basis, leaves the tool in working condition longer, and the quality of the treated surface increases. This way significantly increases productivity and production efficiency. However, its use is often not allowed due to technical conditions. In this case, it is advisable to use tools with carbon diamond-like and diamond coatings.

All coatings were uniformly formed on the working surfaces of the carbide inserts and the radius of rounding of the cutting edge slightly increased; it was 42–45 μm on the original plate and 50–55 μm on the coated plates.

The cutting conditions provided a surface roughness no worse than Ra = 0.8 microns. When the plate was worn out by more than 0.3 mm on the back surface, there was intense aluminum sticking on the cutter, which hindered the processing process and significantly worsened the roughness parameter. Therefore, this wear value was chosen as the criterion for the failure of the cutting plate. Photos of the clips can be seen in Figure 7.

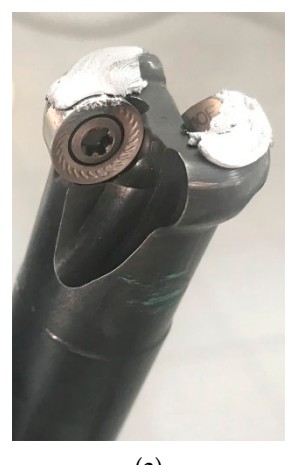 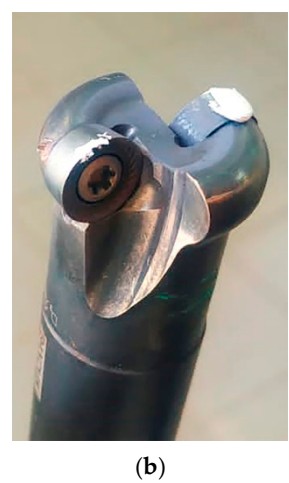 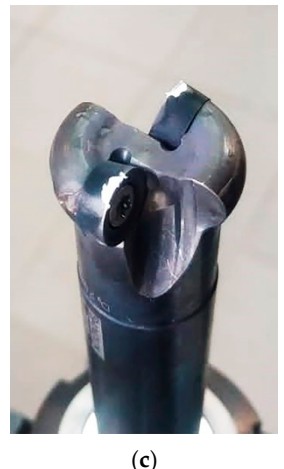 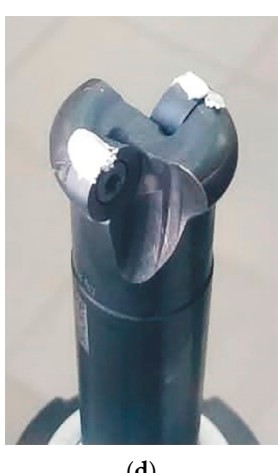

(**a**)          (**b**)          (**c**)          (**d**)

**Figure 7.** Aluminum sticking on the assembly mill: (**a**) uncoated insert after 20 passes (cutting length L = 8 m), (**b**) ncAlTiCrN/Si3N4 + DLC-coated insert after 40 passes (L = 16 m), (**c**) MCD/NCD-coated insert after 40 passes, and (**d**) after 80 passes (L = 36 m); the insert diameter is 5 mm.

Coating significantly increases the durability of the tool. Figure 8 shows experimentally obtained dependences of the wear value along the flank surface $h_f$ of the cutting plate on the cutting length L. The wear pattern of all the studied plates was classical. The zones

of run in, steady state, and critical wear were traced. For uncoated tools, the milling time before the edge reaches the critical wear value on the back surface was T = 400 s at a cutting speed of 800 m/min, which corresponds to a cutting length of L = 8.5 m; the effect for carbon coatings is markedly different. Thus, the cutting length of the carbide insert with ncATiCrN/$Si_3N_4$ + DLC coating doubled (T = 800 s, L = 17 m), and the plate with PCD coating increased four times when the wear criterion was reached (T = 1600 s, L = 34 m).

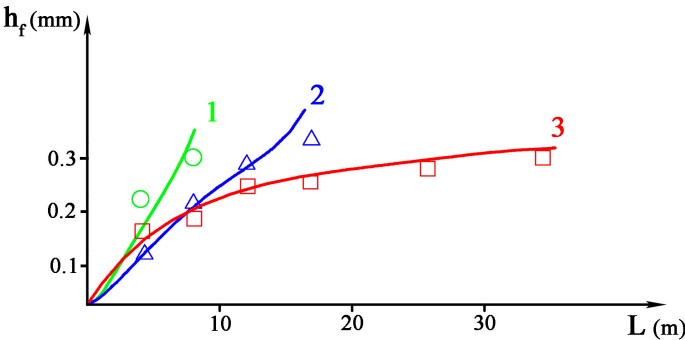

**Figure 8.** The durability of ball-end mills with coatings: 1—uncoated, 2—ncAlTiCrN/$Si_3N_4$ + DLC, 3—MCD/NCD.

Figures 9–11 show 3D images of the cutting edges of the plates before and after wear and their light sections obtained on the multifunctional optical measuring system MikroCAD premium+ (GFM, Teltow, Germany) after etching aluminum deposits in a warm (50 C) 10% solution of potassium hydroxide (KOH) when exposed to ultrasound. Figures 12–14 show the state of the cutting edge after the plate reaches the established wear criterion of 0.3 mm and the distribution of Al and Si on the working surfaces of the cutter.

Several researchers have noted that during high-speed milling of aluminum alloys, the main contribution to the wear of the carbide tool on the rake surface was made by the adhesive–oxidative wear mechanism. At the same time, forming an adhesive layer led to a decrease in the cutting force but increased the width of the wear zone along the flank surface, where only adhesive wear is predominant [38]. However, when processing a composite alloy, due to the high proportion of SiC in the workpiece material, the abrasive wear of the tool is also significantly intensified [14]. Thus, tool wear is a complex result of abrasive and adhesive wear. At the same time, experiments show that the brittle destruction of particles such as SiC plays a crucial role in the specific mechanism of mechanical processing of MMC, which additionally leads to an increase in cutting force, increased wear on the flank surface, and formation of subsurface damages [39].

In our case, it was possible to observe how aluminum was first smeared on the flank surface of the plate. The size of the adhesive increased with increasing wear on the flank surface, which can be attributed to a decrease in the primary relief angle. At the same time, regardless of the presence of the coating, a significant reduction in the value of the radius of rounding of the cutting edge was observed when the insert tip was ground off. In the case of the initial plate, it decreased from 53 to 32 μm (Figure 9c,d), with ncAlTiCrN/$Si_3N_4$ + DLC coating from 53 to 32 μm (Figure 10c,d), with PCD coating from 51 to 21 μm (Figure 11c,d).

On the worn rake and flank surfaces of the original cutting plate, the relief formed by WC grains when removing the cobalt binder (Figure 9b) is visible. This predominant movement of the binding phase lead to the staining of tungsten carbide grains. At the same time, there were no traces of oxidative wear. It can be seen that the wear of the cutting edge of the insert also occurs due to the penetration of SiC hard abrasive particles or their fragments into the hard alloy with the formation of grooves. This mechanism is undoubtedly present, and is visible in Figure 9, with the remaining aluminum not etched in alkali in the corresponding scratches. It is possible to observe stuck silicon carbide particles on the flank surface near the cutting edge (Figure 9d). In this case, there is also a so-called

soft abrasion [40], in which wear occurs due to the extrusion of the cobalt binder between the tungsten carbide grains, followed by their precipitation.

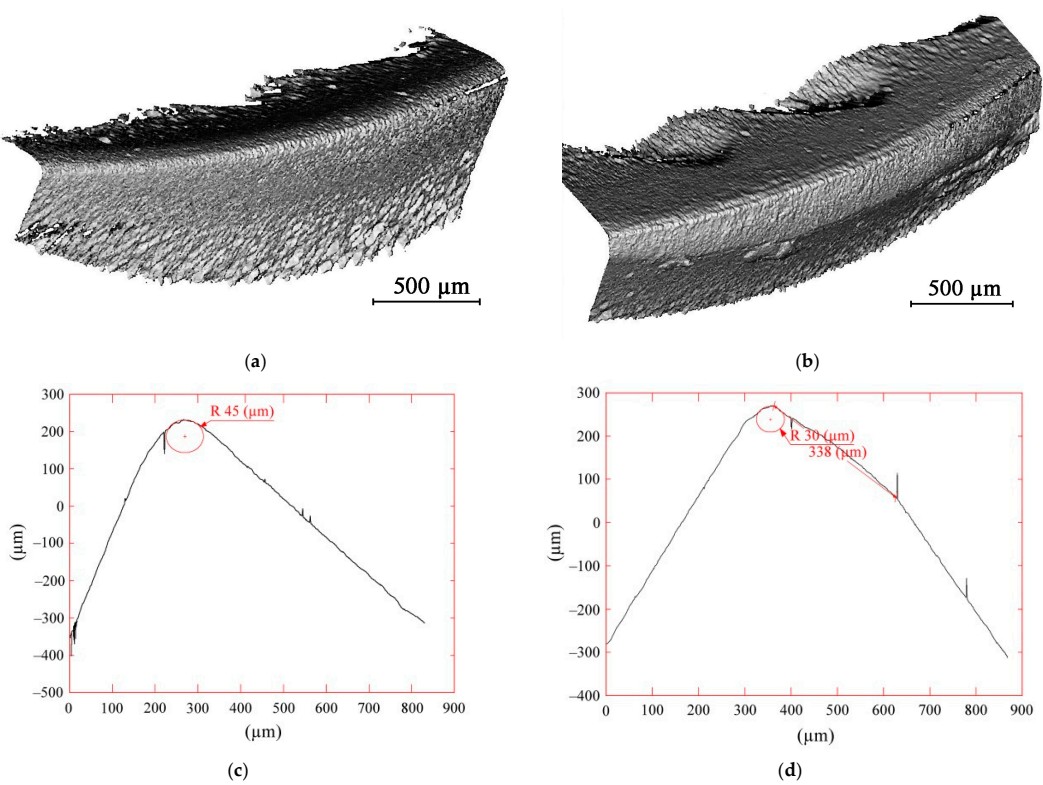

**Figure 9.** The cutting edge of the initial carbide plate before (**a**) and after 20 passes (**b**) and the change in the radius of its rounding on the light section: initial (**c**), after 20 passes (**d**).

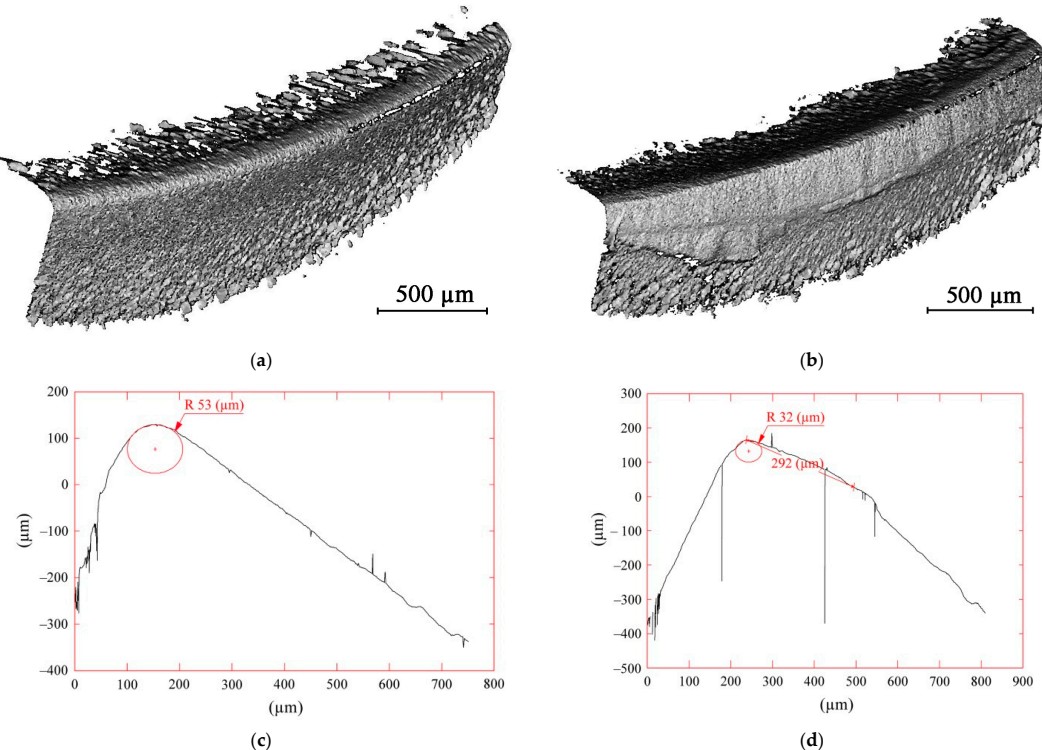

**Figure 10.** The cutting edge of the carbide plate with ncAlTiCrN/Si$_3$N$_4$ + DLC coating and a change in the radius of its rounding on the light section before (**a**,**c**) and after 40 passes by the cutter (**b**,**d**).

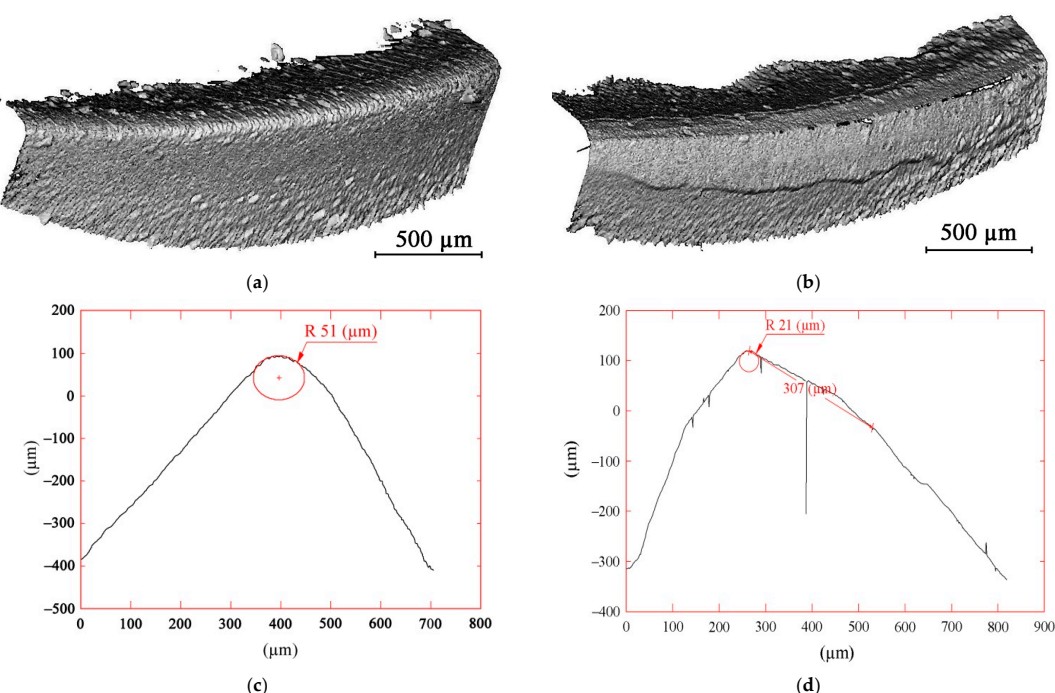

**Figure 11.** The cutting edge of the carbide plate with PCD coating and a change in the radius of its rounding on the light section before (**a**,**c**) and after 80 passes by the cutter (**b**,**d**).

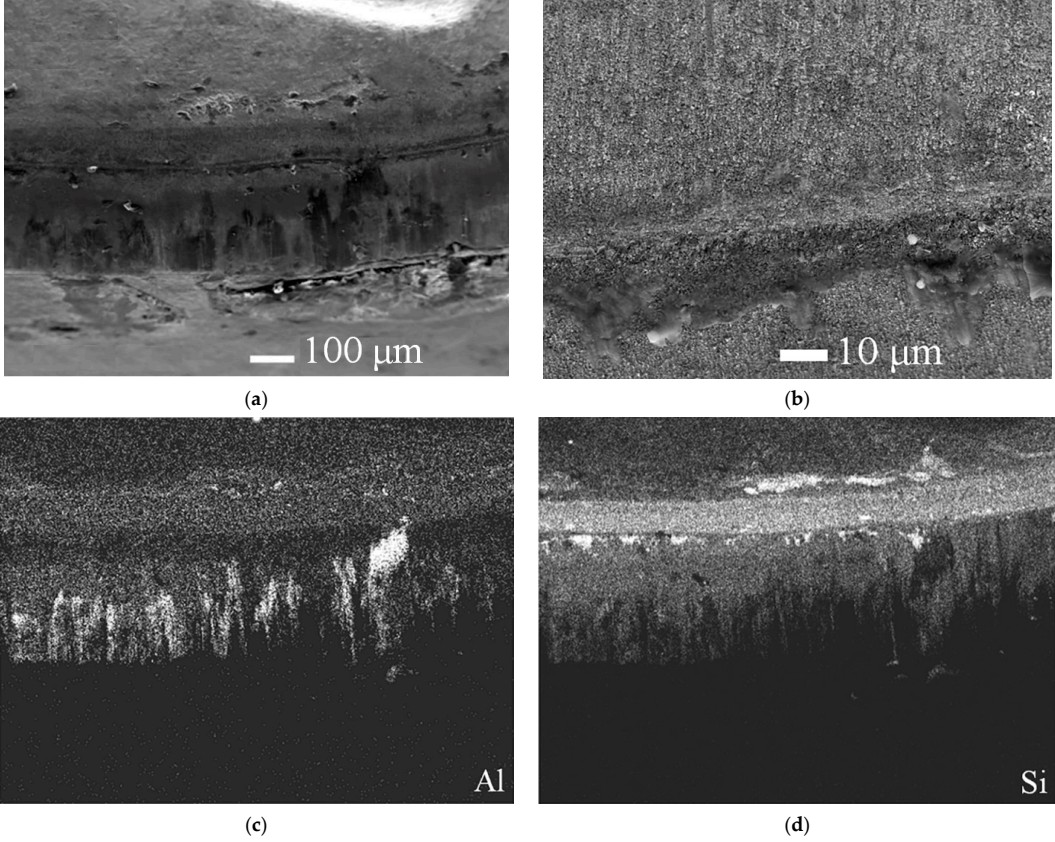

**Figure 12.** (**a**,**b**) SEM image of a worn CoroMill 200 plate uncoated after 20 passes of the cutter (cutting length 8.5 m), (**c**) distribution of Al in the cutting-edge area, (**d**) distribution of Si in the cutting-edge area.

When the ncAlTiCrN/Si$_3$N$_4$ + DLC-coated plate wears out after 40 passes of the mill, the effect of soft abrasion is observed only on the flank surface of the insert and near the cutting age, where a small cavity has formed (Figures 10 and 13). The abrasive wear mechanism prevails on the rake surface. The surface relief is more even, and the carbide grains are not so pronounced. This can be attributed to the fact that the antifriction layer of the diamond-like coating significantly reduces friction on the contact pad of the carbide plate, reduces the likelihood of formation of a hardening when processing abrasive aluminum alloys and thereby slows down the formation of wear chamfers. The role of the ncAlTiCrN sublayer is manifested in the fact that more favorable conditions are created for the functioning of the DLC layer, the strength of its adhesive connection with the tool material increases, the stress level in the coating decreases, and it becomes more stable to withstand thermal and power loads longer [41]. Even after the outer layer of DLC has worn off, the nitride sublayer acts as a wear-resistant coating, slowing down the wear rate.

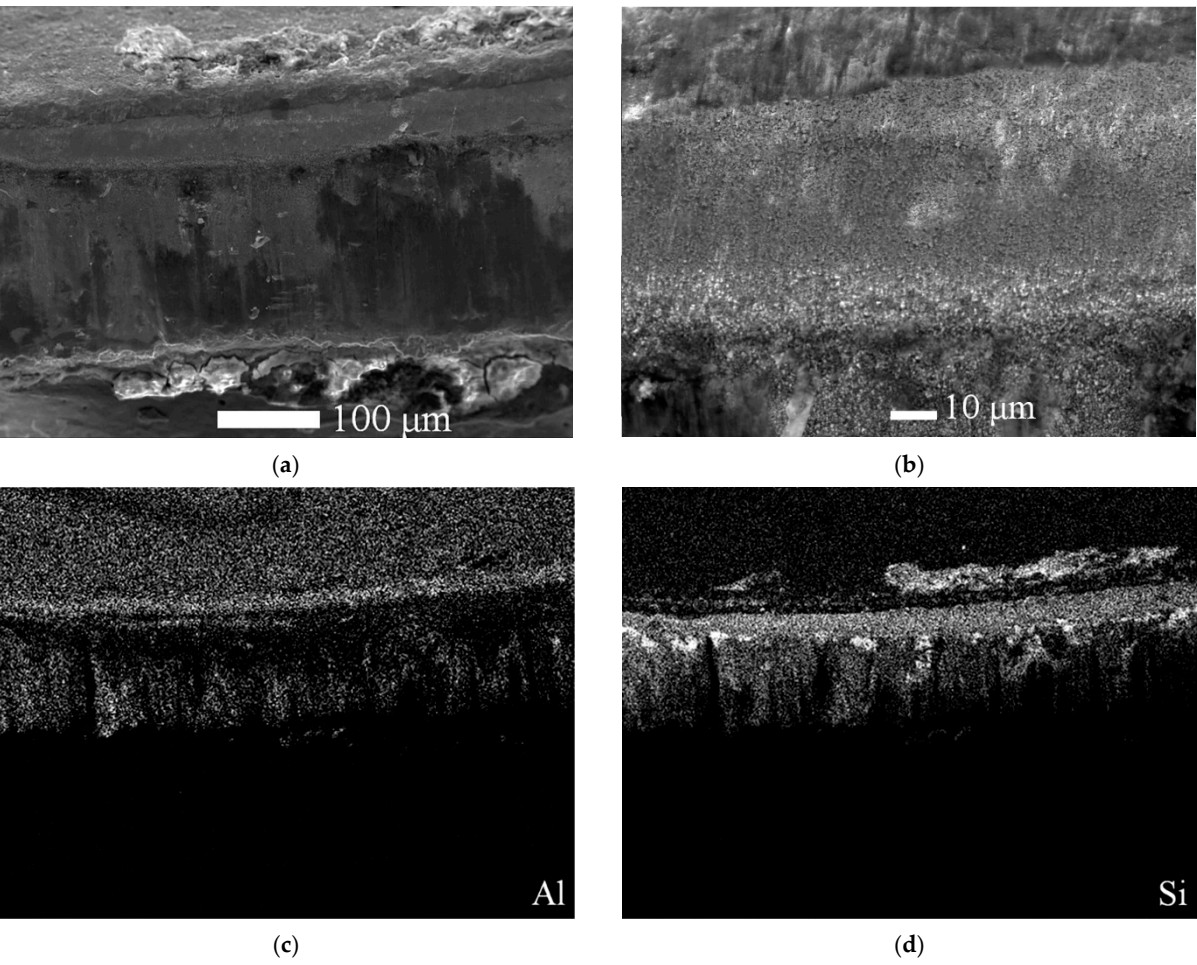

**Figure 13.** (**a**,**b**) SEM image of a worn CoroMill 200 plate with ncAlTiCrN/Si3N4 + DLC coating after 40 passes of the cutter (cutting length 17 m), (**c**) distribution of Al in the cutting-edge area, (**d**) distribution of Si in the cutting-edge area.

In the case of a tool with MCD/NCD coating (Figures 11 and 13), it should be noted that a large wear chamfer appears on the rake surface, which can be observed on the light section of the cutting edge (Figure 13b,d), which can be associated with a long tool operation time. The preliminary preparation of the surface by etching played an essential role in this, which significantly reduced the cobalt content in the near-surface zone. The wear on the flank surface increased quite quickly to a value of 0.15–0.2 mm in about 20 passes of the mill and then stabilized and slowly increased, reaching a value of 0.3 mm only by the 80th pass.

Wear is mainly abrasive. It is possible to observe non-etched remnants of aluminum sticks near the insert cutting age on the rake surface (Figure 14c) and silicon carbide particles on the flank surface in the form of tiny whiskers (Figure 14d). The distribution of carbon over the surface of worn samples obtained by EDS shows that the area of the diamond film preserved after testing is more than 12 times the size of the diamond-like film.

It suggests that adhesive wear is also present but is not the main one here since there are no breaks of the base material from the cutting edge. The mechanism of soft abrasion is wholly suppressed. The high thermal conductivity of the diamond coating plays an essential role here, which significantly reduces the temperature in the cutting-edge area [42].

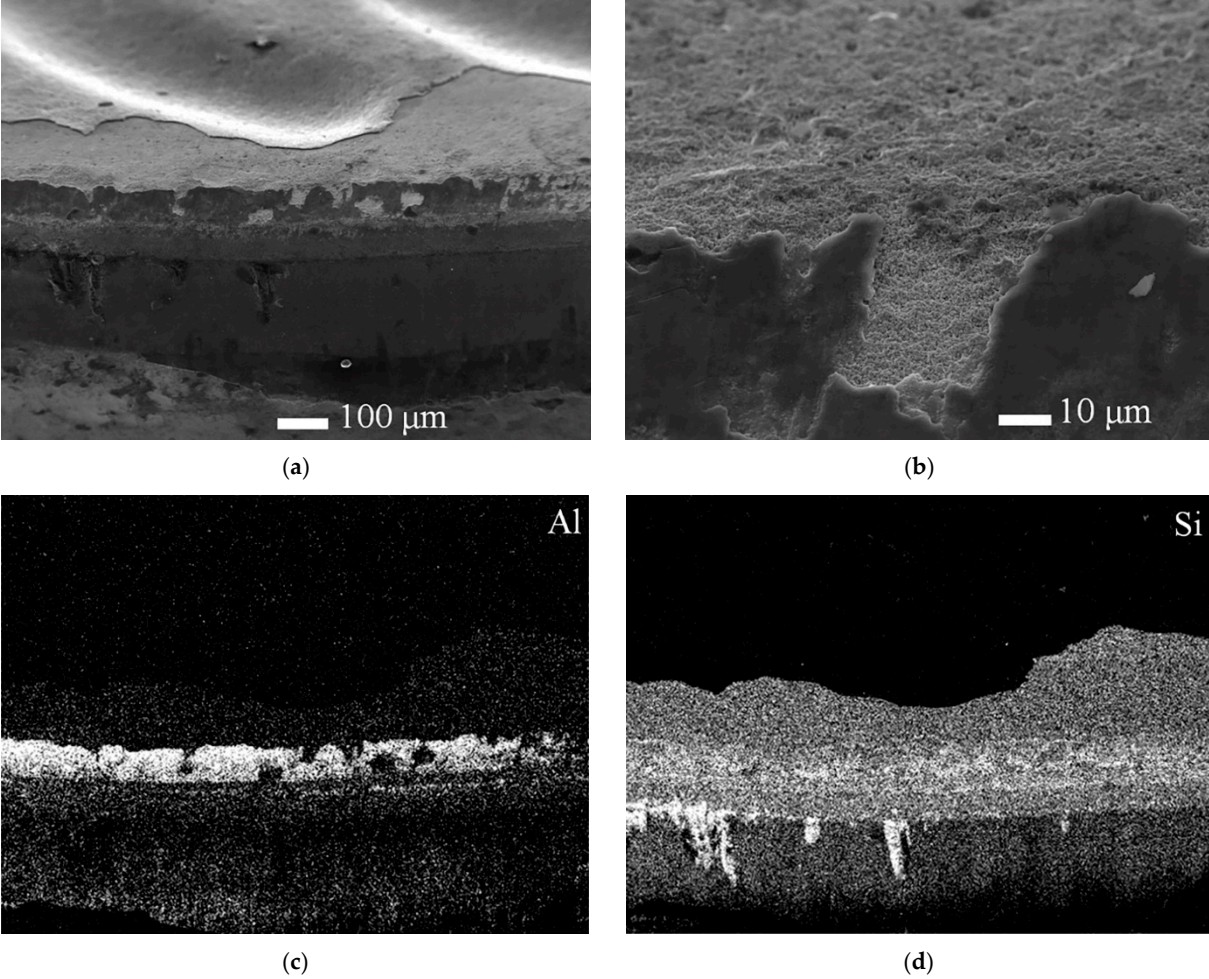

**Figure 14.** (**a**,**b**) SEM image of a worn CoroMill 200 plate with MCD/NCD coating after 80 passes of the cutter (cutting length 34 m), (**c**) distribution of Al in the cutting-edge area, (**d**) distribution of Si in the cutting-edge area.

The low melting point of the aluminum composite is not capable of causing a tool to wear to a high level. However, the service life of the instrument inevitably depends on the cutting process. However, an increase in the temperature and the components of the cutting force to a certain level can lead to a change in the microstructure and the occurrence of a level of residual stresses in the surface layer, reaching the strength limit of the hard alloy. This effect is possible when worn with a hard abrasive, such as SiC. In this case, the material is removed from the wear surface due to plastic deformation due to the malleability and extrusion of the Co binder, followed by a weakening of the compressive stresses in the WC grains and by their cracking and fragmentation. If the diffusion or mechanical coupling between Co and WC is weak, whole grains can be removed from the wear surface. If there

is a strong diffusion bond between the two phases, the material will be removed by brittle peeling because cracks spread through both phases due to abrasive wear.

It can be assumed that in the case of the application of wear-resistant carbon coatings on the tool, the diffusion interaction of Co and WC in the wear zone is enhanced due to the formation of secondary structures on the friction surface that appear as a result of saturation of the carbon phase with cobalt, tungsten, and their oxides [43]. Scratches formed by friction on the coating surface are filled with all types of wear particles and can also be trapped for relatively large separated particles. This process of damage healing by wear particles can prevent the development of fatigue failures in stress concentration zones.

Secondary structures, self-organizing on the surface of coatings in the form of nanoparticle chains distributed in the sliding direction [25], formed on graphite and diamond components of coatings, apparently have a different composition, which affects the nature of wear and the period of tool durability. On graphite-like ingredients, secondary structures probably contain more Co than diamond components, which is manifested in the suppression of the soft abrasion mechanism only on the rake surface, unlike the PCD variant, when the wear mechanism changes more significantly.

## 4. Conclusions

Milling of matrix composite alloys based on aluminum causes a sufficient number of problems, which are caused by the high abrasiveness of the material and its tendency to stick to the cutting surface. A significant achievement of modern MMC processing technologies has been using tools with carbon coatings, diamondlike and polycrystalline diamond, with high hardness and a low coefficient of friction. The conducted studies show that a tool with an NCD/MCD coating can be the most effective, due to its high thermal conductivity, chemical inertia, and wear resistance. It once again confirms that the weak chemical affinity of diamond with aluminum is the predominant factor in improving machinability.

The durability of CoroMill 200 cutting plates with NCD/MCD coating is two times higher than with DLC and four times higher than that of uncoated inserts, with high-speed milling of MMC at a speed of 800 m/min, a cutting depth of 1 mm, and a feed of 0.2 mm/tooth.

The primary mechanism of wear of an uncoated carbide tool under a given cutting mode is soft abrasion, in which wear on both the front and back surfaces occurs due to the extrusion of a cobalt binder between tungsten carbide grains, followed by their loss.

A diamond-like coating can change the wear mechanism of the cutter, but only on the front surface, where the abrasive mechanism is manifested. At the same time, the wear rate slows down.

A diamond multilayer coating based on alternating layers of micro- and nanoscale polycrystals (MCD/NCD) completely changes the wear mechanism, preventing the tungsten carbide grains from staining from the cutting surface of the plate, providing better tool durability with such a coating.

Thus, this article shows the advantage of a diamond-coated tool. However, there is an obvious need for further research concerning the choice of architecture and thickness of diamond films, providing an optimal combination of the critical properties "thermal conductivity–the strength of the adhesive bond with the substrate-microhardness-modulus of elasticity", taking into account the material of the tool and the expected operating conditions.

**Author Contributions:** Conceptualization, V.I.K. and E.E.A.; methodology, A.K.M.; investigation, V.G.R. and A.A.K.; validation, V.S.S. and R.A.K.; formal analysis, M.A.M. and S.G.R.; project administration, S.N.G.; writing and editing S.V.F. All authors have read and agreed to the published version of the manuscript.

**Funding:** This research was carried out at the expense of the grant of the Russian Science Foundation No. 22-19-00694.

**Institutional Review Board Statement:** Not applicable.

**Informed Consent Statement:** Not applicable.

**Data Availability Statement:** The data presented in this study are available on request from the corresponding author.

**Acknowledgments:** This study was carried out on the equipment of the Center of Collective Use of MSUT "STANKIN".

**Conflicts of Interest:** The authors declare no conflict of interest.

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
