# Peer review of "Wear of Carbide Plates with Diamond-like and Micro-Nano Polycrystalline Diamond Coatings during Interrupted Cutting of Composite Alloy Al/SiC"

_jmmp, doi:10.3390/jmmp7060224_

Round 1
Reviewer 1 Report
Comments and Suggestions for Authors
1. In the abstract, please add the results in terms of percentage.
2. Introduction section should include the latest references. The last paragraph of the introduction should include the contribution in bullet points. The experimental setup (Fig.1) should include a comprehensive figure and label it. Fig.2 should be labeled. Combine Fig. 1 to Fig. 5 in a single figure. SEM figures should be labeled. Show tool life graph and explain it.
3. It would be helpful to briefly elaborate on the significance of addressing tool wear in this specific context and how it aligns with the broader field of materials processing.
4. The authors' choice of modifying tool surfaces through plasma chemical deposition of carbon-based multilayer coatings (DLC or PCD) is innovative and potentially transformative in enhancing tool reliability. It would be beneficial if the authors could provide a concise rationale for choosing this approach over alternative methods in results and discussion. In this way, it would be easy to see the differences between techniques.
5. The use of an indexable mill with CoroMill 200 inserts in the experiments adds practical relevance to the study. However, a more detailed description of the experimental setup, such as the conditions and parameters considered, would enhance the reproducibility of the findings.
7. The identification of the primary wear mechanism as soft abrasion, with the extrusion of the cobalt binder between tungsten carbide grains, is well-explained. It would be helpful if the authors could briefly discuss the implications of this mechanism on the overall tool performance and potential implications for practical applications.
8. The differentiation between abrasive wear and soft abrasion, along with the limited presence of adhesive wear, is a strong point. However, further clarification on how these findings contribute to understanding the longevity of the coated tools in different wear scenarios would be beneficial.
9. The observation that abrasive wear begins to prevail against the background of soft abrasion is intriguing. It would be insightful if the authors could discuss the potential reasons behind this transition and its implications for the long-term performance of the coated tools.
10. The conclusion effectively summarizes the key findings. To enhance the completeness of the review, the authors might consider briefly suggesting potential avenues for further research, such as exploring variations in coating thickness or composition under different machining conditions.
Need to engage a native English speaker to revise the language.
Author Response
see file

Reviewer 2 Report
Comments and Suggestions for Authors
p.5/figure 3 Can the authors label the cobalt elements ?
p.8/figure 7 From the picture you can see the stuck material particles on the cutting plate. Nothing is described in the text about the creation of an increase. Can the authors comment on this?
p.9/r.257 „Figures 10,12,14 shows the state of the cutting edge after the plate reaches the established wear criterion of 0.3 mm and the distribution of some chemical elements on the working surfaces of the cutter.“
From this description, I don't know which is figure 14. Which chemical elements are visible in the figures 10, 12, 14 ?
p.9/r.280 „There are no traces of oxidation, which suggests that the temperature in the cutting zone was less than 700 °C.“ is based on what?
p.9/r.250 Figure 8 shows the values ​​of the cutting length in mm, while in the experiments the machined surface was in m. The criterion was reached for curve 3 because it is not clear from Figure 8?
p.11/r.293 and p.12/r.309 two figures 10 are marked
p.13/r.322, 323 Figure 14c,d - Which is Figure 14 ?
p13/r.325 Missing a more detailed description of the figures : 15a a 15b
p.15/r.350..... 800 m/s or p.4/r.133............ 800 m/min What is right ?
For the defined wear criterion, we obtain the durability value. What then does the statement express „high wear resistance“ (p15/r.347) ?
The article is interesting for theory and practice and I recommend it for publication in a journal after corrections.
Mainly, it is necessary to harmonize the numbering of the Figures from the description in the text, and to describe some Fixtures better.
In the case of some claims, if they are not the result of research, I recommend not to state or add a reference to the literature.
Author Response
see file

Reviewer 3 Report
Comments and Suggestions for Authors
1. Composite of SiCp/Al must be mentioned in the abstract and keywords.
2. The unit in figure 5 is incorrect.
3. What does hf refer to in figure 8. It is not mentioned.
4. The labels in figure 9 is needed. And what is the meaning of cutter (i, c)
5. Two figure 10s are presented. And what is mkm in figure 10(a)
6. 'back surface' can be replaced by 'flank surface', and 'front surface' by 'rake surface
7. Figure 14 is omitted.
Author Response
see file

Reviewer 4 Report
Comments and Suggestions for Authors
Introduction
1. On page 2 line 85. “The higher the hardness of the processed alloys, the lower the roughness of their surface due to the reduction of sticking to the tool.” Is there any literature support this statement?
2. The gap between literature and this paper is not clear, therefore the introduction section is lack of the novelty of the work.
Major body of the manuscript
3. In “2.2. Tool coatings” section, it is suggested to create a table showing all the coating properties (such as chemical composition, thickness, technique, etc.)
4. On page 7 line 217. There is typo for “Figure 6. Raman spectra of CVD and DLC coating samples at a distance of 0.5 mm from the cutting age.”
5. On page 8 line 242. Figure 7. Please convert 20/40/80 passes into meters.
6. On page 10 line 290-291. Figure 9 title is confusing.
7. On page 11 and page 12. There are two “Figure 10”. Also, please fix the scale “mkm” on (a). What are (c) and (d) for? There is no explanation in the text.
8. The figures in the manuscript are mixed and disordered. This makes confusing when matching the text and explanation. There is no scale on some of the figures, which are important. Please fix all the figures.
9. The explanation for Results and Discussion section is lack of experiment support. Only SEM images showing wear patterns is not strong enough to explain the scientific reason. More measurements should be conducted.
Comments on the Quality of English Language
10. Some spelling and formatting errors need to be checked.
Author Response
see file

Round 2
Reviewer 2 Report
Comments and Suggestions for Authors
Recommend to publish without fundamental comments.
Author Response
Dear reviewer!
Thank you so much for your work!
S. Fedorov.
Reviewer 3 Report
Comments and Suggestions for Authors
The revision is OK!
Author Response
Dear reviewer!
Thank you so much for your work!
S. Fedorov
Reviewer 4 Report
Comments and Suggestions for Authors
There are still some minor issues, such as on page 16 line 388, coating "NCD/NCD". It is not clear which coating is that.
The authors should carefully check the whole content regarding terminology issues.
Comments on the Quality of English Languageminor mistakes
Author Response
Dear reviewer!
Thank you so much for your work!
The terminology has been checked, errors has been corrected.
S. Fedorov